# Coping mechanisms for students with mental disorders: An exploratory qualitative study at Busitema University's Mbale and Busia Campuses

Enid Kawala Kagoya [1] *, Joseph Mpagi[2], Paul Waako[3], Martha R. L. Muhwezi[4], Agnes Namaganda [5], Allan G. Nsubuga [1], Christine Etoko Atala[6], Francis Okello[1], Ambrose Okibure[1], Ronald Kibuuka[1], Ernest Wandera[1], Kalisiti Ndamanywa[2], Joseph Kirabira[7]

1 Department of Community Health, Institute of Public Health, Faculty of Health Sciences, Busitema University, Mbale, Uganda, 2 Department of Academics, Research and Innovation, Faculty of Health Sciences, Busitema University, Mbale, Uganda, 3 Busitema University, Busia, Uganda, 4 Forum for Africa Women Educationalists (FAWE) Africa, Kampala, Uganda, 5 Department of Physiology, College of Health Sciences, Makerere University, Kampala, Uganda, 6 Department of Anesthesia, Mbarara University of Science and Technology, Mbarara, Uganda, 7 Department of Psychiatry, Faculty of Health Sciences, Busitema University, Mbale, Uganda

* enidkawala@gmail.com

## Abstract

In recent years, Ugandan universities have faced a rising incidence of mental health issues among students, with prevalence rates of mental disorders reaching up to 60% among undergraduates. These challenges significantly impact both academic performance and social interactions. This study aimed to investigate the coping mechanisms among students with mental disorders at Busitema University. We conducted 42 key informant interviews with students diagnosed with mental disorders, as determined by the Mini International Neuropsychiatric Interview. Interviews were guided by a structured protocol developed by the research team, and all participants provided informed consent. The interviews were audio-taped, transcribed verbatim, and analyzed using thematic analysis. This approach was chosen for its efficiency and directness. Nvivo software facilitated the coding and organization of themes. The study identified several coping strategies used by students at the two Busitema University campuses. Five main themes emerged: Alcohol and Substance Use, Use of Sedatives, Social and Personal Initiatives, Seeking Counseling Services, Withdrawal and Confrontation, and Faith Healing. Various subthemes were also noted, including substance use, withdrawal, confrontations, witchcraft, participation in co-curricular activities, abortions, and seeking psychosocial support. These findings are detailed in Table 2. Addressing the mental health challenges faced by university students requires a thorough understanding of their coping strategies. While some strategies are self-developed, others involve university-led interventions. It is crucial to reinforce effective coping mechanisms and address detrimental ones to improve students' mental health outcomes. Mental disorders, coping strategies, university students, Uganda

**Data Availability Statement:** All relevant data are within the paper and its Supporting Information files.

**Funding:** The study received minimal funding from the University due to the urgency of the phenomenon that we explored, and the funds were mainly used to facilitate data collection. The funders (BURIFs) grant no. 3/DGSRI/22 had no role in study design, data collection and analysis, decision to publish, or preparation of the manuscript. None of the investigators received salary from the funders. JK, a senior lecturer in the department of Psychiatry was responsible for the small money that facilitated the data collection exercise.

**Competing interests:** The authors have declared that no competing interests exist.

## Introduction

The World Health Organization (WHO) defines health as a state of complete physical, mental, and social well-being, not merely the absence of disease or infirmity. In this context, mental well-being is crucial as it significantly impacts overall productivity [1]. Mental health, an integral component of overall health, includes intellectual, spiritual, and emotional dimensions [2]. The World Psychiatric Association (WPA) outlines four key standards of mental health: physical and mental coordination, adaptation to the social environment, well-being, and the ability to fully utilize one's potential at work [3].

In Uganda, many students in higher education institutions face mental health challenges, yet systematic coping strategies are often lacking. Instead, individuals create their own strategies based on their unique problems and manifestations [4]. The WHO highlights that university students need a stable mood, coordinated interpersonal relations, objective self-assessment, and psychological adaptability [5]. The transition from adolescence to adulthood and moving away from home to attend university can increase the risk of depression, anxiety, and stress, raising concerns about the effectiveness of students' coping mechanisms [6].

Understanding these coping strategies is essential for designing interventions to address the high prevalence of mental health issues among students [7]. A WHO survey found that over 30% of college students in eight countries experienced mental distress, which adversely affected their academic performance and daily lives, leading to role impairments [8]. Students typically employ various strategies to manage mental health issues, including religious support [9], positive reinterpretation [10], active coping [11], planning, and instrumental support [12]. Additional methods may involve breathing exercises [12], regular counselling sessions, talking to someone, temporary distractions, social networking, and frequent exercise, healthy eating, meditation, joining clubs, mindfulness, and calming techniques [5].

While some students exhibit resilience and effective coping [1], others may resort to disengagement, isolation, over-indulgence, grief, and internalized coping strategies, which can negatively impact physical health and exacerbate mental health conditions [13]. Mental health remains a peripheral issue in health policy and development agendas, with insufficient strategies to create supportive environments for optimal mental health [12]. This is reflected in the lack of comprehensive data on the extent of mental health issues and coping strategies among undergraduate [1]. Stigmatization of mental illness, combined with limited innovative psychosocial interventions and counselling resources, as well as the inadequate capacity to deliver mental health services, contributes to students' hesitancy in seeking help and their negative attitudes toward counselling services [11].

Given the limited research on undergraduate students' coping strategies for mental health challenges, our study aims to explore and understand these strategies among students at Busitema University's Tororo and Mbale campuses. We will investigate the coping mechanisms adopted by students and the support systems available to address their mental health needs.

## Materials and methods

### Study designs

This was a descriptive qualitative study among students at Busitema University faculties of Health Sciences and Engineering.

### Study site

The study was conducted at two sites within Busitema University: the Faculty of Health Sciences at the Mbale Campus and the Faculty of Engineering at the main Busitema Campus. The

Faculty of Health Sciences is situated within the Mbale Regional Referral Hospital in Eastern Uganda, serving patients from 16 districts. This faculty primarily offers undergraduate and postgraduate medical courses, including Bachelor's degrees in Medicine and Surgery, Nursing, and Anesthesia. The Faculty of Engineering, located at the main Busitema Campus in Tororo District, focuses on Engineering courses at the diploma and bachelor's levels, including Mechanical and Civil Engineering.

## Participants

The study included undergraduate students from Busitema University's Faculty of Health Sciences and Faculty of Engineering who were diagnosed with one or more mental disorders using the Mini International Neuropsychiatric Interview (MINI). Students with severe or emergency medical conditions that prevented meaningful participation in the interviews were excluded from the study.

## Data collection, instruments, and analysis

A total of 42 key informant interviews were conducted with students diagnosed with mental disorders. Participants were approached during their free time through their student leaders. Purposive sampling was employed, initially screening students for mental disorders such as depression, anxiety, bipolar disorder, and psychotic disorders using the MINI version 7.0.2. Students who tested positive for any of these disorders were selected for qualitative interviews based on the severity and number of disorders. Those who consented to participate were scheduled for interviews. A written informed consent was sought form all participants. The interviews were guided by a semi-structured interview guide developed by the research team. To develop semi-structured interviews, we began by defining the research goals and identifying key themes to be explored. A literature review was done to help us identify gaps and refine the areas of focus. We then created an interview guide with open-ended questions and follow-up prompts based on these themes, ensuring flexibility for deeper exploration. A pilot test with a small sample allowed us to refine the guide for clarity and relevance. Throughout, we maintained ethical standards, including confidentiality and informed consent. This approach allowed for both focused questioning and flexibility for participants to provide additional information. All interviews were audiotaped, transcribed verbatim, and reviewed multiple times by the research team. Thematic analysis was conducted following an inductive approach guided by specific objectives with Nvivo software facilitating the coding and organization of data. The process involved several steps: The initial coding was performed by EK and JK, responsible for identifying and labelling significant data segments. While NVivo software was utilized to support the organization and management of codes, the initial coding was done manually by the researchers. NVivo was instrumental in storing, organizing, and analyzing the coded data. Data entry into NVivo was carried out by EKK responsible for data entry. This involved inputting verbatim transcripts and organizing them within the NVivo environment for subsequent analysis. Codes were derived through an iterative process involving multiple readings of the transcripts. The research team developed codes based on recurring themes, patterns, and significant insights identified during the readings. Themes were extrapolated by grouping and analyzing the coded data to identify overarching patterns and concepts. This process involved constant comparison and refinement to ensure that the themes accurately reflected the data. The identified codes and themes were reviewed and validated by EKK involved in validation, ensuring reliability and consistency in the analysis. During thematic analysis the team first became familiar with the data, generated initial codes, searched for themes, reviewed themes, defined themes and started writing them up. We enhanced

reliability through inter-coder reliability checks, where multiple coders reviewed and discussed coding discrepancies to reach a consensus. An audit trail was maintained to document the coding decisions and thematic development process. The study ensured validity by triangulating data sources, using member checking, and maintaining a detailed audit trail. These measures helped confirm that the findings were grounded in the data and accurately represented participants' perspectives. A comprehensive audit trail was kept throughout the research process, documenting all decisions related to data collection, coding, and theme development. This audit trail provided transparency and allows for the replication of the analysis process.

## Ethical considerations

Ethical approval was obtained from the Research Ethics Committees (REC) of Busitema University (Faculty of Health Sciences, BUFHS-2022-11) and the National Council for Science and Technology (UNSCT-Number: HS2700ES). Interviews were conducted in secure and private settings to ensure confidentiality and written informed consent was sought from all participants. The study included students who whose written consent was obtained during the study. The study did not include minors. Administrative clearance for data collection was granted by the Vice Chancellor's office. All research procedures adhered to national and international guidelines for conducting research with human participants, especially during the COVID-19 pandemic.

## Results

A total of 42 students participated in the interviews, comprising 9 males and 11 females from the Faculty of Health Sciences, and 15 males and 7 females from the Faculty of Engineering (see Table 1).

The participants reported coping mechanisms that predominantly fell into five thematic areas: alcohol and substance use, social and personal initiatives, seeking counselling services, withdrawal or disengagement, and faith healing (see Table 2).

### Theme 1: Alcohol and substance use as a coping mechanism

The study identified alcohol and substance use as prevalent coping strategies among undergraduate students at Busitema University's Mbale and Tororo campuses. Many students admitted to using beer as a way to cope with their problems, while some also used other substances such as marijuana and cannabis. Others opted for cheaper alternatives, such as "coke" (a local soft drink), although a few could afford to drink coffee regularly. Additionally, some students reported using sedatives to address sleep-related challenges associated with their mental health disorders.

**Subtheme 1: Use of alcoholic beverages.** Students reported consuming alcoholic beverages as a coping mechanism, with some drinking beer on weekends and others daily after a demanding day. At the Mbale campus, students noted that the intense academic schedules and pressures led them to use wines, beers, and other alcoholic drinks to alleviate stress and recharge for the next day

**Table 1. Social demographics of participants.**

| Faculty | Male | Female | 1st year | 2nd year | 3rd year | 4th Year | 5th year |
|---|---|---|---|---|---|---|---|
| **Health Sciences** | 9 | 11 | 5 | 5 | 6 | 1 | 3 20 |
| **Engineering** | 1 | 7 | 6 | 4 | 6 | 6 | 22 |
| **Total** | 23 | 20 | 11 | 9 | 12 | 7 | 3 |

**Table 2. Thematic areas of coping mechanisms among university students.**

| Theme | Subtheme |
|---|---|
| **Alcohol and substance** | Use of alcoholic drinks like beers, Waragi |
| | Use of substances like Marijuana, cannabis, marijuana coffee, sedatives and others. |
| | Commonly used sedatives |
| | Other sedatives |
| **Social and personal initiatives** | Reading Novels and other books |
| | Social gatherings like parties |
| | Engaging in curricular activities |
| | Opportunities, confrontation and abortion. |
| **Seeking counselling services** | Peer-led/ parental/ Church-related counselling |
| | Professional counselling from university counsellors |
| **Withdrawal or confrontation** | Social Media |
| | Parents and relatives |
| | Academic stressors |
| | Relationships |
| | Isolation |
| **Faith healing** | Prayer and fasting |
| | Meditations |
| | Witchcraft |

"*I take beer, especially after a stressful day, we meet at some place in Shawole with some colleagues and have some kind of fun.*

*(FOE, Male, Year 3, BCT, 27 years)*

"*I take wines, Smirnoff, but once in a while, Like at social events like parties, when I get a chance to go to clubs etc*

*(FHS, Female, Year 5, MBchB, 24 years)*".

**Subtheme 2: Substance use among students.** The study revealed that some students admitted to using illicit drugs and narcotics. They reported that these substances helped them "forget their hardships," provided a feeling of euphoria, and, importantly, were perceived to enhance their memory retention, thereby aiding their studies

"*. . . . . .some of us have been taking tramadol and others smoke weed and other stimulants because we think that when you smoke it, we shall forget your problems, you don't think of that anymore*

*(FHS, Male, Year 3, MBchB, 23 years)*"

"*. . . . . .Some of us smoke a lot and take alcohol for example I take a tusker just to keep myself calm and smoke some weed but once in a while with my friends. . ..*

*(FOE, Male, Year 4, WAR, 26 years).*"

"*Yes, I used to use a lot of coffee and without it, I thought I would not function, feel sad in class and tired all the time. If I failed to get coffee, I would get a Coca-Cola or Pepsi because I knew they had caffeine and it would make me energetic though it brought me problems*

*because I took it in high amounts and took me back to the depression" crying. . . . . . . ..2 minutes*

*(FHS, Female, Year 5, MBchB, 25years)."*.

**Sub-theme 3: Use of sedatives as a coping mechanism for students with mental health challenges.** After a long working day, some students opted for sleeping pills and other sedatives to help them relax and get some rest. The study identifies the types of sedatives used and the contexts in which they were employed by the students.

*Subtheme 3a*: *Use of sleeping pills*. Several students reported using sleeping pills to manage the intense pressure, particularly those in their clinical years. These students preferred sleeping pills as a means to alleviate stress and obtain much-needed rest amidst the demanding university workload.

"*I took an overdose of diazepam (I took a strip of diazepam) and the next day I found myself partially dead. Some people broke into my room and rushed me to the hospital and gave me an antidote and other treatment. I have been using it for a while to catch my sleep after trying other options and they disappointed me,*

*(FHS, Male, Year 3, MBchB, 23 years)."*

"*. . . . .One time I was forced to take sleeping pills that's when I was rotating in internal medicine and ob-gyn, the pressure was too much and I had to look for an alternative,*

*(FHS, Female, Year 5, MBchB, 26 years)"*.

"I used to take sleeping peels of diphenhydramine" all the time I wanted to have constructive sleep but woke up the following day when my friends were coming back from class

*(FHS, Female, Year 5, MBchB, 24 years)"*.,

*Subtheme 3b*: *Other sedatives*. The study also revealed that some students used other substances, such as coffee and excessive amounts of Coke, as sedatives. These substances were employed to either boost their energy levels or, conversely, to slow down their activity and reduce their attention span.

"*I would also take amitriptyline, I know it was a bad coping mechanism but it was one way to go,*

*(FHS, Female, Year 4, BNS, 25 years)*

"*There is a ka drug I bought but I overslept and missed lectures that day and vowed not to take it again because I thought that when I slept for long the anxiety would go but it didn't, I felt better for some time and went back. I got it over the counter from one of the pharmacies around town.*

*(FOE, Male, Year 2, AMI, 23 years)*

## Theme 2: Social and personal initiatives

The study also revealed that students developed personal initiatives to maintain their mental stability. These initiatives included reading novels, attending social events such as parties and club activities, and participating in extracurricular activities.

**Subtheme 1: Reading novels and other books.** Students who chose to read novels reported doing so to alleviate anxiety and depression. Some indicated that, during periods of suicidal thoughts, reading the Bible provided them with a valuable source of comfort and relief.

"*I do read novels most times when I feel life has lost meaning and I encourage other students to do the same instead of crying whenever they are depressed.*

*(FOE, Female, Year 3, BEE, 26 years)*"

"*There are times when I had financial and family issues, I felt like killing myself but I resorted to reading some good chapters in the bible and realized that Jesus loves me, killing myself was not an option*

*(FHS, Male, Year 2, MBchB, 2 years).*

**Subtheme 2: Attending social gatherings like parties.** Attending social gatherings, such as parties, nightclubs, and movie nights, emerged as a significant coping mechanism among students at both campuses. Many students reported that dancing the night away at clubs helped them relieve stress and pressure. Additionally, participating in events like medical dinners, cultural galas, and weddings of peers provided a respite from university-related pressures and contributed to their mental well-being

"*Once in a while, I go to Club el-kanji when there are artists, I dance and kill the stress off as I wait for the next week and its problems*

*(FHS, Female, Year 1, BNA, 22 years)*".

"I attend parties for our chapel and colleagues who are getting married which keeps me off my final year project for some time

*(FOE, Male, Year4, BCT, 26 years)*"

"*We have a cultural gala in Mbale which I participate in and during that whole period, I forget about the stress of books, exams etc.*

*(FHS, Male, Year 3, BNS, 21 years)*"

**Subtheme 3: Engaging in curricular activities.** The study found that students at the Tororo campus frequently engaged in co-curricular activities as a coping mechanism, in contrast to those at the Mbale campus, where limited space constrained participation. At Tororo campus, students utilized these activities to address their mental health challenges effectively.

"*Yes, I play football with friends every evening and this has helped me to avoid being alone and overthinking*

*(FOE, Male, Year DGE, 24 years)*".

"*Yeah at our compass, we don't have where to play football from however we have had some time to play at Mbale College of Health Sciences, and this has made us relieved a little*

*(FHS, Male, Year 4, BNA, 26 years)*".

**Subtheme 5: Opportunities, confrontation and abortion.** Confrontation emerged as a mechanism that students believed could help resolve problems, particularly misunderstandings with their lecturers.

"*For the Lecturer who segregates us in class, my dear. . .. eh. . ..I went ahead and confronted that lecturer who felt like his tribe mates were more important than some of us, I know I did a wrong thing but the guy styled*

(**FHS**, **Male**, **Year 5**, **MBchB**, **24 years)**".

Some students, particularly those who had lost their jobs and were the sole income providers for themselves and their families, believed in and made optimal use of available opportunities for a better future.

"*I am looking for an alternative job in Mbale and I got one in Abacus but it was in Kampala and yet it was hard for me to balance books and school. When I got another one, the opportunity was in Mbale and Arua and they would not allow me to choose the place mate site they ended up posting me in Arua and I missed out just like that*

(**FHS**, **Male**, **Year 2**, **BNA**, **30 Years)**".

Regarding accommodation, some students chose to rent apartments around town despite feeling insecure. Due to the poor conditions of their hall of residence, they opted to pair up and pay monthly rent, which was costly but considered preferable to the hostels.

"*We ended up going out as in renting but we did not have that good security like at Bellodian and the rent was too much for us. I have friends who were robbed because they are renting and even within bellodian that kept traumatizing them*

(**FHS**, **Female**, **Year 5**, **MBchB**, **24 years)**".

To address financial challenges, some students sought parental support, applied for scholarships from various organizations, or organized fundraising campaigns to assist peers facing financial difficulties. Additionally, students who were not enrolled full-time actively sought job opportunities to provide financial support.

"*My colleagues ran a fundraising campaign for me and we were in a position to raise 850 K and for sure when they gave me that money, it was enough for me to heal psychologically, I felt like it was better than nothing*

(**FHS**, **Female**, **Year 1**, **BNS**, **21 years)**".

"*I got a sponsor through a small organization called "Blue House" in Kazo district and it took me up and we stayed there from P.5 up to S. 6. Life was okay but it would take a toll on your emotional well-being*

(**FHS**, **Female**, **Year 2**, **MBChB 22 years)**".

".*I was forced to borrow money from colleagues and money lenders who have also become a headache to me*

(**FOE**, **Male**, **Year 2**, **MEB**, **24 years)**".

**Subtheme 6: Listening to music and socializing.** The study also revealed that listening to music was a common method for students to manage psychological distress. Additionally, some students kept themselves busy as a way to distract from the challenges and stresses of their daily lives.

"*I listen to Christian music all the time as long as I am not at school and it has helped me to deal with the suicidal behaviour which I had*

*(FOE, Male, Year 1, AMI, 20 years)*".

"*I am a very task-oriented person, so being in the ward would make me forget some of the things related to the problems I had,*

*(FHS, Female, Year 4, MBchB, 28 years)*

## Theme 3: Seeking counselling services

While some students employed the aforementioned coping strategies, counselling emerged as another widely embraced method at both campuses, despite a shortage of professional counsellors. To address this gap, students utilized alternatives such as peer-led counselling, parental support, and visits to professional counsellors and psychiatrists for additional assistance.

**Subtheme 1: Peer-led/ parental/ Church-related counseling.** Seeking counselling from peers, parents, and church leaders was identified as a coping strategy for some students. They felt more comfortable sharing their problems with these individuals, reflecting the adage, "A problem shared is a problem halved."

"*I just kept talking to my friends and trying the above alternatives for the medical issues but for counselling, it was for my issues,*

*(FHS, Female, Year 5, MBchB, 30 years)*".

"*I would also talk to my parents and they offered all the necessary support they got me some mentors and they kept with me all the time,*

*(FHS, Male, Year 3, BNA, 21 years)*

"*I trusted my friends and parents especially when I felt like I was developing bipolar. I also got self-help from U-Tube channels,*

*(FHS, Female, Year 1, BNS, 20years,)*"

**Subtheme 2: Professional counselling from university counselors.** Some students preferred to seek help from professional counsellors and psychiatrists. While some were referred to these professionals, others sought out their services independently for psychosocial support.

"*I also talked to the Psychiatrist and the university Psychologist who gave me some talk therapy,*

*(FHS, BNA, Year 3, 32 years)*"

"*Yeah, I talked to some but not my problems, I would tell them about academic issues because even when we used to tell them, they used to tell us that it's the nature of medical school and you have to persevere,*

*(FHS, Female, Year 4, BNS, 27 years)*

### Theme 4: Withdrawal as a coping strategy

Some students chose to withdraw from certain aspects of their lives such as relationships, social media, and nagging relatives or friends to focus more on their academics and personal goals, which they believed improved their mental well-being.

**Subtheme 1: Withdrawal from social media.**   The study revealed that some students decided to disengage from social media, which had previously contributed to their mental health challenges. They found that exposure to distressing content and messages on social media often left them feeling depressed, and withdrawing from these platforms became a necessary solution for alleviating their mental distress.

"*I withdrew from all social media platforms and switched off my phones and we went for training and distanced myself from things that were taking a lot of my thoughts*

*(FHS, Female, Year 3, MBchB, 23 years).*

"*I turned off social media and withdrew which used to expose me to some funny messages and also changed companies as in the people I was associating with from people*

*(FOE, Male, Year 2, WAR, 23 years).*

**Subtheme 2: Withdrawal from parents and relatives.**   Some students also chose to distance themselves from certain relatives and friends who had become sources of stress and negatively impacted their mental well-being.

"*I distanced myself from people whom I thought had negative thoughts towards me*

*(FOE, Female, Year 4, WAR, 24 years).*

"*I avoid talking to people and that is why you can find me with heads on even while moving on the streets just to avoid talking to people.*

*(FHS, Female, Year 3, MBchB, 23 years)*".

**Subtheme 3: Withdrawal from academic stressors.**   Withdrawing from leadership positions and other academic stressors was also a coping strategy employed by some students at both campuses.

"*Students ended up leaving the course and others while in year one decided to apply and join other universities because the administration was not supportive,*

*(FOE, Male, Year 2, DCE, 21 years)*".

"*I had many leadership roles like class representative, choir leader, Rotaract treasurer, I was in women in the medical world etc. but I withdrew from them and my life took another direction. At that time my BP normalized because it was unstable and I used to get palpitations*

*(FHS, Male, Year 5, MBchB, 25 years)*"

**Subtheme 4: Withdrawal from relationships.**   For some students, withdrawing from relationships was a personal choice to escape associated complications and relieve stress. They found purpose in focusing on their academic goals, ambitions, and pursuits.

"*For relationships, I decided to concentrate on the person I wanted to be and also withdrew from the relationships though you can see me pregnant I won't go for any other relationship, this was a lesson to me.*

*(FOE, Female, Yeah 4, WAR, 26 years).*

"*I withdrew from the relationship I had with some lecturer, went to church and prayed for the guy, My worry was a retake in his course unit but I read hard and passed*

*(FHS Female, Year 3, MBchB 23 years)*".

**Subtheme 5: Isolation.** Due to the stigma associated with mental illness, some students at both campuses primarily adapted by isolating themselves, secluding themselves from others, and avoiding social contact.

"*When you let people know what you are going through, they will think you are mad. I just keep to myself lonely and avoid listening to people that's why you see me with headsets all the time*

*(FHS, Female, Year 3, MBchB, 23 years)*".

## Theme 5: Faith healing

The study also identified seeking help from supernatural sources as a coping strategy, including prayer and fasting, meditation, and witchcraft. It was noteworthy that some students at the Tororo campus turned to witchcraft to revive their businesses despite fierce competition, and also used it as a means to restore and maintain relationships.

**Subtheme 1: Prayer and fasting.** The study revealed that many students turned to prayer and fasting as primary coping strategies amid their psychological distress and mental health challenges. They found solace in biblical principles such as Philippians 4:5–7, which encourages believers to present their requests to God through prayer and supplication, promising peace that surpasses understanding, and Matthew 17:21, which emphasizes that certain challenges are overcome through prayer and fasting.

The study identified another coping strategy which was seeking help from supernatural sources through prayer and fasting, meditation and witchcraft. It was surprising that some of the students at Tororo Compass preferred to use witchcraft to revive their business despite the tight competition around them. Witchcraft was also identified as a solution to the restoration and maintenance of relationships by some of the students.

"*Sometimes I would go to the chapel for prayers and praise and worship and also engage in some days of prayer and fasting,*

*(FHS, Female, Year 1, MBchB, 21 years)*".

"*I talked to my friends, especially those we missed, cried over it and gave up because we had it in mind that the lecturer just wanted to punish us, so I cried and cried and gave up.*

*(FHS, Female, Year 5, MBchB, 24 years")*.

"*I used to pray a lot and talk to God and it could make me better with both depression and anxiety.*

*(FOE, Male, DAG, 26 years)*"

**Subtheme 2: Meditation.** Some students actively engaged in meditation, which helped them maintain stable mental health.

"*I went through guided meditation and learnt new habits of controlling myself and read books "No one can affect your moods unless you have accepted" so things became okay*

*(FOE, Male, Year 2, AMI, 21 years).*

"*I was going through a brainwash, I could get something and meditate for 30 minutes and avoid thoughts that would be running in my mind*

*(FHS, Male, Year 2, BNA,21 years).*

**Subtheme 3: Witchcraft.** The study also revealed that some students at the Tororo campus engaged in witchcraft. It was surprising to find that, despite intense competition, these students used witchcraft to revive their businesses. Additionally, witchcraft was employed by some as a means to restore and maintain personal relationships.

"*When my business was not progressing well and it was my source of tuition, I was advised by colleagues to try witchcraft since most of the colleagues in business were doing so and it worked for me, I cleared all my tuition*

*(FOE, Male, Year 2, BEE, 22 years).*

"*I was forced to use witchcraft because I was told that it was the only remedy that would return my girlfriend. I had already done kukyala and waited for us to finish school and have a family but that guy snatched her and it gave me sleepless nights for almost a week,*

*(FOE, Male, Year 3, WAR 26 years*".

"*Here in Shawole, most people use witchcraft in such scenarios, I also gave it a try though no results yet*

*(FOE, Male, Year 4 BCT, 26 years*".

## Discussion

This study aimed to explore the primary coping strategies among undergraduate students at Busitema University who were diagnosed with mental health disorders. Our findings revealed that while students at both campuses used a variety of coping strategies, some common approaches included faith healing, substance use, and withdrawal.

The use of alcohol and other substances was notably linked to heavy workloads, exam pressures, final-year projects, and inadequate learning environments. In addition to substance use, students utilized various social, economic, and personal coping strategies. Some found relief from academic stress through substance use, while others engaged in personal activities such as reading novels and listening to music. Our findings are consistent with studies conducted among students in countries like Nigeria and Ghana, where mental health challenges are similarly addressed through a mix of personal and social coping strategies [14]. Many students with mental health issues faced significant stigma, often being perceived as cursed, mad, or possessed. This stigma led to suffering in silence and isolation due to the spiritual connotations associated with mental illness. Consequently, those affected frequently sought divine intervention through various religious institutions, such as churches and mosques. Similarly, in our

study, seeking spiritual intervention emerged as a prevalent coping strategy among students at both campuses, aligning with findings from a study conducted in Turkey [11]. Although witchcraft and curses are commonly believed to cause mental illness and general misfortunes, some students at one of the campuses turned to these practices to address financial and relationship problems. This usage reflects the broader cultural belief in witchcraft as a solution to various personal issues [15]. Religious support has been shown to positively influence hope, life satisfaction, and emotional well-being. This finding is consistent with a study conducted in Poland during the COVID-19 pandemic, which also highlighted the benefits of religious support in enhancing psychological resilience [7]. Although the use of alcohol and other substances is generally discouraged, [6] students at both campuses frequently used stimulants and amphetamines, such as tramadol and cannabis, particularly marijuana, to manage stress. This pattern is consistent with findings from Kamran Sattah's scoping review, which also noted the prevalence of such substances among students [2]. Our findings also indicated that receiving psychosocial support through peer-led, parental, and professional counsellors was an effective coping strategy for reducing the prevalence of mental disorders among students at both campuses. This aligns with similar findings from a study conducted among college students in Bangladesh [10]. Providing guidance and counselling services at both campuses is a commendable initiative that students would likely benefit from. However, the effectiveness of these services was undermined by issues related to trust, confidentiality, and a shortage of counsellors, particularly female counsellors, at both campuses. Additionally, many students had a negative attitude toward counselling, which further impacted its uptake and efficacy [16], creating awareness on guidance [1] and counselling and employing trained counselors [17] could enhance access to counselling services by students. Mental health education should be introduced [12] to all students at all campuses of Busitema University and cascaded to other institutions as well.

## Strengths and limitations

This study effectively captured responses from students diagnosed with mental disorders at both campuses. However, a limitation was the constrained timeframe allocated for the study.

## Conclusion

Mental health challenges among students are prevalent in higher education institutions. Understanding and improving coping strategies both self-devised and university led is crucial for enhancing students' mental health. Streamlining interventions and increasing mental health support are necessary for better mental health outcomes.

## Acknowledgments

This study was fully supported by the Busitema University Research and Innovation Fund grant 3/DGSRI/22. We extend our gratitude to the administration of Busitema University and the student leaders at both the Busitema and Mbale campuses for their support throughout the data collection process.

## Author Contributions

**Conceptualization:** Enid Kawala Kagoya, Joseph Kirabira.

**Data curation:** Ernest Wandera.

**Formal analysis:** Enid Kawala Kagoya, Allan G. Nsubuga.

**Methodology:** Enid Kawala Kagoya.

**Supervision:** Enid Kawala Kagoya.

**Validation:** Agnes Namaganda.

**Writing – original draft:** Enid Kawala Kagoya.

**Writing – review & editing:** Joseph Mpagi, Paul Waako, Martha R. L. Muhwezi, Allan G. Nsubuga, Christine Etoko Atala, Francis Okello, Ambrose Okibure, Ronald Kibuuka, Kalisiti Ndamanywa.

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
