## [Decision Letter · Decision Letter 0]

13 Jul 2023

PGPH-D-23-01027

Coping mechanisms of students with mental disorders at Busitema University, an exploratory qualitative study at Mbale and Busia campuses.

Dear Dr. Kawala,

Thank you for submitting your manuscript to PLOS Global Public Health. After careful consideration, we feel that it has merit but does not fully meet PLOS Global Public Health’s publication criteria as it currently stands. Therefore, we invite you to submit a revised version of the manuscript that addresses the points raised during the review process.

We look forward to receiving your revised manuscript.

Kind regards,

Anish Veshnal Cherian, Ph.D.

Academic Editor

Journal Requirements:

1. Please provide additional details regarding participant consent. In the ethics statement in the Methods and online submission information, please ensure that you have specified (1) whether consent was informed and (2) what type you obtained (for instance, written or verbal, and if verbal, how it was documented and witnessed). If your study included minors, state whether you obtained consent from parents or guardians. If the need for consent was waived by the ethics committee, please include this information.

3. Please amend your detailed Financial Disclosure statement. This is published with the article. It must therefore be completed in full sentences and contain the exact wording you wish to be published.

4. Please provide separate figure files in .tif or .eps format only and remove any figures embedded in your manuscript file. Please also ensure all files are under our size limit of 10MB.

Additional Editor Comments (if provided):

Reviewers' comments:

Reviewer's Responses to Questions

**Comments to the Author**

1. Does this manuscript meet PLOS Global Public Health’s publication criteria? Is the manuscript technically sound, and do the data support the conclusions? The manuscript must describe methodologically and ethically rigorous research with conclusions that are appropriately drawn based on the data presented.

Reviewer #1: Yes

Reviewer #2: Partly

2. Has the statistical analysis been performed appropriately and rigorously?

Reviewer #1: Yes

Reviewer #2: N/A

3. Have the authors made all data underlying the findings in their manuscript fully available (please refer to the Data Availability Statement at the start of the manuscript PDF file)?

Reviewer #1: Yes

Reviewer #2: No

4. Is the manuscript presented in an intelligible fashion and written in standard English?

Reviewer #1: Yes

Reviewer #2: No

5. Review Comments to the Author

Reviewer #1: This is a useful manuscript but I would request the authors to make some changes to improve the scientific rigor.

1. How did the authors decide the number of participants? Were they targeting data saturation? Authors need to clarify this.

2. If the analysis was deductive, did the authors decide a prior what the themes were? Could the authors clarify this?

3. There are several limitations. As this is a qualitative study, it's not certain whether the results are generalizable. The findings are from one medical college and one engineering college. The findings need to be verified using quantitative methods. The authors have used MINI. Which version MINI? MINI does not pick up all psychiatric disorders. So the authors cannot generalize their findings to everyone with a psychiatric disorder.

4. Could the authors use the checklists for qualitative studies to improve the rigor?

5. Could the authors provide clarifications for use of key informants for this study?

6. Could the authors provide details of the semi-structured guide?

Reviewer #2: The study is a good initiative, however, require major revision.

1. Title of the study should be relooked. The study states that they had recruited students who had positive diagnosis according to MINI. However, a major part of the results exactly do not depict that the coping strategies used by the students are to mitigate this said 'Mental Disorders', rather their psychological distress due to academic stress, financial stress or relationship issues, which essentially may not qualify to be classified as mental disorders. So the study fails to tell us in the discussion or in te conclusion how this coping mechanism is different for a person with mental disorder, while compared to a person without mental disorder or how the MINI assessment was significant for the study.

2. The method used is sound and 42 interviews is absolutely a great sample. However, a major revision is need in the themes as the themes and title don't match exactly. For example:

2.a. In the theme, opportunity, confrontation and abortion, the verbatim actually depicted the challenges faced by the students rather than their coping mechanism.

2.b. In the theme Withdrawal, the theme withdrawal from relationships, the reviewer fails to understand whether the relationship had anything to do with their mental disorder or how they used it as a coping mechanism to for their mental disorder.

2.c. The sub-theme ,use of witchcraft, again was to alleviate their financial struggles or difficulties with relationship, or to be successful in business, but not of mental health issues. None of the verbatim supported the idea that the respondents used witchcraft to reduce their concerns related to mental disorder/mental health concerns unlike the usual pathway to care of using witchcraft to reduce the distress due to the symptoms of mental disorders.

2.d. The sub-theme, Using alcohol, the second verbatim says about social drinking, but not a coping mechanism.

3. The study should be reworked grammatically including punctuations, spellings etc. For example in the short title it is mentioned Copying instead of Coping, throughout the text, Campus is misspelt as Compass, Peels instead of Pills.

4. While considering the ethical part of the study, the students are positively diagnosed for psychiatric disorders. Were they already on treatment? If not, when the research team, assessed them for psychiatric disorders, did they refer them to appropriate treatment facility? What was the SOP when you interviewed the respondents with suicidal ideations?

6. PLOS authors have the option to publish the peer review history of their article (what does this mean?). If published, this will include your full peer review and any attached files.

**Do you want your identity to be public for this peer review?** For information about this choice, including consent withdrawal, please see our Privacy Policy.

Reviewer #1: **Yes: **priya sreedaran

Reviewer #2: No

---

## [Decision Letter · Decision Letter 1]

29 Mar 2024

PGPH-D-23-01027R1

Coping mechanisms of students with mental disorders at Busitema University, an exploratory qualitative study at Mbale and Busia campuses.

Dear Dr. Kawala,

Thank you for submitting your manuscript to PLOS Global Public Health. After careful consideration, we feel that it has merit but does not fully meet PLOS Global Public Health’s publication criteria as it currently stands. Therefore, we invite you to submit a revised version of the manuscript that addresses the points raised during the review process.

We look forward to receiving your revised manuscript.

Kind regards,

Miquel Vall-llosera Camps

Staff Editor

Journal Requirements:

1. Please provide additional details regarding participant consent. In the ethics statement in the Methods and online submission information, please ensure that you have specified (1) whether consent was informed and (2) what type you obtained (for instance, written or verbal, and if verbal, how it was documented and witnessed). If your study included minors, state whether you obtained consent from parents or guardians. If the need for consent was waived by the ethics committee, please include this information.

Additional Editor Comments:

Please carefully address the reviewer's remaining scientific concerns about your study.

Reviewers' comments:

Reviewer's Responses to Questions

**Comments to the Author**

1. If the authors have adequately addressed your comments raised in a previous round of review and you feel that this manuscript is now acceptable for publication, you may indicate that here to bypass the “Comments to the Author” section, enter your conflict of interest statement in the “Confidential to Editor” section, and submit your "Accept" recommendation.

Reviewer #1: All comments have been addressed

2. Does this manuscript meet PLOS Global Public Health’s publication criteria? Is the manuscript technically sound, and do the data support the conclusions? The manuscript must describe methodologically and ethically rigorous research with conclusions that are appropriately drawn based on the data presented.

Reviewer #1: Yes

3. Has the statistical analysis been performed appropriately and rigorously?

Reviewer #1: Yes

4. Have the authors made all data underlying the findings in their manuscript fully available (please refer to the Data Availability Statement at the start of the manuscript PDF file)?

Reviewer #1: Yes

5. Is the manuscript presented in an intelligible fashion and written in standard English?

Reviewer #1: Yes

6. Review Comments to the Author

Reviewer #1: Kindly note the following:

1. Qualitative study manuscripts need to have clear details on their methodology and tools. The authors have used a semi-structured guide. It is advisable for authors to mention the basis of formulating this guide. This will make the study more useful for those who are unable to appreciate the contextual factors responsible for the guide.

2. The authors need to mention who did the entry into NVIVO. Also, while NVIVO helps with coding and categorization but the eventual thematic analysis is done by the study authors. The authors need to specify this. The authors should remember that in a qualitative study, the study methods need to be generalizable while the results are often not generalizable.

3. With respect to deductive analysis, this refers to analysis or coding done on basis of pre-existing themes etc. Again, as mentioned already, the authors need to understand that other readers of this manuscript might not have access to the reasoning for the pre-decided codes. I would urge the authors to mention the basis of deductive reasoning.

4. STROBE guidelines are NOT the guidelines for Qualitative. I have attached the relevant link.

https://www.equator-network.org/reporting-guidelines/srqr/. The authors can also use COREQ.

7. PLOS authors have the option to publish the peer review history of their article (what does this mean?). If published, this will include your full peer review and any attached files.

**Do you want your identity to be public for this peer review?** For information about this choice, including consent withdrawal, please see our Privacy Policy.

Reviewer #1: **Yes: **PRIYA SREEDARAN

---

## [Decision Letter · Decision Letter 2]

10 Jun 2024

PGPH-D-23-01027R2

Coping mechanisms of students with mental disorders at Busitema University, an exploratory qualitative study at Mbale and Busia campuses.

Dear Dr. Kawala,

Thank you for submitting your manuscript to PLOS Global Public Health. After careful consideration, we feel that it has merit but does not fully meet PLOS Global Public Health’s publication criteria as it currently stands. Therefore, we invite you to submit a revised version of the manuscript that addresses the points raised during the review process.

We look forward to receiving your revised manuscript.

Kind regards,

Abhijit Nadkarni

Academic Editor

Journal Requirements:

1. Please provide additional details regarding participant consent. In the ethics statement in the Methods and online submission information, please ensure that you have specified (1) whether consent was informed and (2) what type you obtained (for instance, written or verbal, and if verbal, how it was documented and witnessed). If your study included minors, state whether you obtained consent from parents or guardians. If the need for consent was waived by the ethics committee, please include this information.

Additional Editor Comments (if provided):

Reviewers' comments:

Reviewer's Responses to Questions

**Comments to the Author**

1. If the authors have adequately addressed your comments raised in a previous round of review and you feel that this manuscript is now acceptable for publication, you may indicate that here to bypass the “Comments to the Author” section, enter your conflict of interest statement in the “Confidential to Editor” section, and submit your "Accept" recommendation.

Reviewer #1: (No Response)

2. Does this manuscript meet PLOS Global Public Health’s publication criteria? Is the manuscript technically sound, and do the data support the conclusions? The manuscript must describe methodologically and ethically rigorous research with conclusions that are appropriately drawn based on the data presented.

Reviewer #1: No

3. Has the statistical analysis been performed appropriately and rigorously?

Reviewer #1: Yes

4. Have the authors made all data underlying the findings in their manuscript fully available (please refer to the Data Availability Statement at the start of the manuscript PDF file)?

Reviewer #1: Yes

5. Is the manuscript presented in an intelligible fashion and written in standard English?

Reviewer #1: No

6. Review Comments to the Author

Reviewer #1: In the version of manuscript that I reviewed, there are still issues to be addressed.

1. The authors should specify that they obtained informed consent in the main draft and not just in the responses to the reviewers.

2. The authors have not mentioned on what basis was the deductive approach used? Did the authors perform a review of literature to obtain the basis of deductive approach? Was it the authors' experience? This needs to be mentioned

3. I would urge the authors to go through this link https://www.equator-network.org/reporting-guidelines-study-design/qualitative-research/?post_type=eq_guidelines&eq_guideli

The authors have put in a lot of effort and it is essential that their manuscript is a good reflection of their efforts. The current version does not do justice to their efforts/

7. PLOS authors have the option to publish the peer review history of their article (what does this mean?). If published, this will include your full peer review and any attached files.

**Do you want your identity to be public for this peer review?** For information about this choice, including consent withdrawal, please see our Privacy Policy.

Reviewer #1: No

---

## [Decision Letter · Decision Letter 3]

6 Aug 2024

PGPH-D-23-01027R3

Coping mechanisms of students with mental disorders at Busitema University, an exploratory qualitative study at Mbale and Busia campuses.

Dear Dr. Kawala,

Thank you for submitting your manuscript to PLOS Global Public Health. After careful consideration, we feel that it has merit but does not fully meet PLOS Global Public Health’s publication criteria as it currently stands. Therefore, we invite you to submit a revised version of the manuscript that addresses the points raised during the review process.

Please note that the reviewer has consistently requested for a detailed description of methodology. Kindly insure that your revision addresses this feedback and contains adequate detail about methodology as is consistent with reporting of qualitative research.

We look forward to receiving your revised manuscript.

Kind regards,

Abhijit Nadkarni

Academic Editor

Journal Requirements:

1. Please provide additional details regarding participant consent. In the ethics statement in the Methods and online submission information, please ensure that you have specified (1) whether consent was informed and (2) what type you obtained (for instance, written or verbal, and if verbal, how it was documented and witnessed). If your study included minors, state whether you obtained consent from parents or guardians. If the need for consent was waived by the ethics committee, please include this information.

Additional Editor Comments (if provided):

Please note that the reviewer's comments have consistently been about provide more details about methodology. Kindly ensure that appropriate details are provided as one would expect with reporting of qualitative research.

Reviewers' comments:

Reviewer's Responses to Questions

**Comments to the Author**

1. If the authors have adequately addressed your comments raised in a previous round of review and you feel that this manuscript is now acceptable for publication, you may indicate that here to bypass the “Comments to the Author” section, enter your conflict of interest statement in the “Confidential to Editor” section, and submit your "Accept" recommendation.

Reviewer #1: (No Response)

2. Does this manuscript meet PLOS Global Public Health’s publication criteria? Is the manuscript technically sound, and do the data support the conclusions? The manuscript must describe methodologically and ethically rigorous research with conclusions that are appropriately drawn based on the data presented.

Reviewer #1: Partly

3. Has the statistical analysis been performed appropriately and rigorously?

Reviewer #1: I don't know

4. Have the authors made all data underlying the findings in their manuscript fully available (please refer to the Data Availability Statement at the start of the manuscript PDF file)?

Reviewer #1: Yes

5. Is the manuscript presented in an intelligible fashion and written in standard English?

Reviewer #1: No

6. Review Comments to the Author

Reviewer #1: The authors have sent a very interesting manuscript.

However, there is a need for rigor in this submission before it is suitable for publication. In qualitative studies, replicability of methods is an important aspect. The authors have not conveyed their qualitative analysis clearly. They have mentioned thematic analysis and deductive approach. However, if the authors were to read the approaches to thematic analysis, they would realize the current description is not enough. They need to specify who did the coding, did they depend entirely on NVivo, who did the entry into NVivo, how did they derive codes and how did they extrapolate themes from these? This discussion on coding is missing. Did the authors use a more reflexive thematic approach or a codebook type of approach?

These questions need to be answered.

7. PLOS authors have the option to publish the peer review history of their article (what does this mean?). If published, this will include your full peer review and any attached files.

**Do you want your identity to be public for this peer review?** For information about this choice, including consent withdrawal, please see our Privacy Policy.

Reviewer #1: No

---

## [Decision Letter · Decision Letter 4]

1 Oct 2024

PGPH-D-23-01027R4

Coping Mechanisms for Students with Mental Disorders: An Exploratory Qualitative Study at Busitema University’s Mbale and Busia Campuses

Dear Dr. Kawala,

Thank you for submitting your manuscript to PLOS Global Public Health. After careful consideration, we feel that it has merit but does not fully meet PLOS Global Public Health’s publication criteria as it currently stands. Therefore, we invite you to submit a revised version of the manuscript that addresses the points raised during the review process.

We look forward to receiving your revised manuscript.

Kind regards,

Abhijit Nadkarni

Academic Editor

Journal Requirements:

1. Please provide additional details regarding participant consent. In the ethics statement in the Methods and online submission information, please ensure that you have specified (1) whether consent was informed and (2) what type you obtained (for instance, written or verbal, and if verbal, how it was documented and witnessed). If your study included minors, state whether you obtained consent from parents or guardians. If the need for consent was waived by the ethics committee, please include this information.

Additional Editor Comments (if provided):

Reviewers' comments:

Reviewer's Responses to Questions

**Comments to the Author**

1. If the authors have adequately addressed your comments raised in a previous round of review and you feel that this manuscript is now acceptable for publication, you may indicate that here to bypass the “Comments to the Author” section, enter your conflict of interest statement in the “Confidential to Editor” section, and submit your "Accept" recommendation.

Reviewer #1: (No Response)

2. Does this manuscript meet PLOS Global Public Health’s publication criteria? Is the manuscript technically sound, and do the data support the conclusions? The manuscript must describe methodologically and ethically rigorous research with conclusions that are appropriately drawn based on the data presented.

Reviewer #1: Partly

3. Has the statistical analysis been performed appropriately and rigorously?

Reviewer #1: N/A

4. Have the authors made all data underlying the findings in their manuscript fully available (please refer to the Data Availability Statement at the start of the manuscript PDF file)?

Reviewer #1: Yes

5. Is the manuscript presented in an intelligible fashion and written in standard English?

Reviewer #1: No

6. Review Comments to the Author

Reviewer #1: While the authors have addressed most of the issues, kindly note the following clarifications and changes required:

1. Which version of MINI did the authors use?

2. How did the authors develop the semi-structured interviews?

3. What were the initial domains anticipated as part of the deductive analysis? Did the generated themes meet these domains?

7. PLOS authors have the option to publish the peer review history of their article (what does this mean?). If published, this will include your full peer review and any attached files.

**Do you want your identity to be public for this peer review?** For information about this choice, including consent withdrawal, please see our Privacy Policy.

Reviewer #1: No

---

## [Editor Report · Decision Letter 5]

3 Nov 2024

PGPH-D-23-01027R5

Coping Mechanisms for Students with Mental Disorders: An Exploratory Qualitative Study at Busitema University’s Mbale and Busia Campuses

Dear Dr. Kagoya,

Thank you for submitting your manuscript to PLOS Global Public Health. After careful consideration, we feel that it has merit but does not fully meet PLOS Global Public Health’s publication criteria as it currently stands. Therefore, we invite you to submit a revised version of the manuscript that addresses the points raised during the review process.

EDITOR:

You have not addressed all the feedback by the reviewer. Please ensure you address all the comments as follows

1. Which version of MINI did the authors use?

2. How did the authors develop the semi-structured interviews?

3. What were the initial domains anticipated as part of the deductive analysis? Did the generated themes meet these domains?

When you submit the revised version ensure the following

1. Only the most recent revisions are highlighted and not the previous revisions

2. Only upload a response document which lists your response to these most recent comments/feedback and not the previous ones

We look forward to receiving your revised manuscript.

Kind regards,

Abhijit Nadkarni

Academic Editor
---

## [Editor Report · Decision Letter 6]

12 Nov 2024

PGPH-D-23-01027R6

Coping Mechanisms for Students with Mental Disorders: An Exploratory Qualitative Study at Busitema University’s Mbale and Busia Campuses

Dear Dr. Kagoya,

Thank you for submitting your manuscript to PLOS Global Public Health. After careful consideration, we feel that it has merit but does not fully meet PLOS Global Public Health’s publication criteria as it currently stands. Therefore, we invite you to submit a revised version of the manuscript that addresses the points raised during the review process.

Thank you for revising your manuscript based on the final comments from the reviewer.

However as indicated in my previous email please submit only the most recent versions of the paper. There are still multiple versions in your submission.

Similarly only upload a response document which lists your response to the most recent comments/feedback and not the previous ones. Currently your submission has multiple tables outlining the revisions you made based on previous feedback

We look forward to receiving your revised manuscript.

Kind regards,

Abhijit Nadkarni

Academic Editor

Journal Requirements:

1. Please provide additional details regarding participant consent. In the ethics statement in the Methods and online submission information, please ensure that you have specified (1) whether consent was informed and (2) what type you obtained (for instance, written or verbal, and if verbal, how it was documented and witnessed). If your study included minors, state whether you obtained consent from parents or guardians. If the need for consent was waived by the ethics committee, please include this information.
---

## [Editor Report · Decision Letter 7]

21 Nov 2024

PGPH-D-23-01027R7

Coping Mechanisms for Students with Mental Disorders: An Exploratory Qualitative Study at Busitema University’s Mbale and Busia Campuses

Dear Dr. Kagoya,

Thank you for submitting your manuscript to PLOS Global Public Health. After careful consideration, we feel that it has merit but does not fully meet PLOS Global Public Health’s publication criteria as it currently stands. Therefore, we invite you to submit a revised version of the manuscript that addresses the points raised during the review process.

However you have still not responded to the feedback from the reveiwer as follows

While the authors have addressed most of the issues, kindly note the following clarifications and changes required:

1. Which version of MINI did the authors use?

2. How did the authors develop the semi-structured interviews?

3. What were the initial domains anticipated as part of the deductive analysis? Did the generated themes meet these domains?

Please respond to these queries by making highlighted changes in the manuscript and uploading a response document which clearly outlines your response to this feedback.

Please do not upload previous versions of the manuscript and documents with responses to previous feedback.

I would really want to approve your manuscript for publication but pending these changes I am unable to move it forward.

We look forward to receiving your revised manuscript.

Kind regards,

Abhijit Nadkarni

Academic Editor
---

## [Editor Report · Decision Letter 8]

26 Nov 2024

PGPH-D-23-01027R8

Coping Mechanisms for Students with Mental Disorders: An Exploratory Qualitative Study at Busitema University’s Mbale and Busia Campuses

Dear Dr. Kagoya,

Thank you for submitting your manuscript to PLOS Global Public Health. After careful consideration, we feel that it has merit but does not fully meet PLOS Global Public Health’s publication criteria as it currently stands. Therefore, we invite you to submit a revised version of the manuscript that addresses the points raised during the review process.

Thank you for responding to the queries and making changes to the manuscript in response to two of the queries. You have provided a detailed response to the third query (*How did the authors develop the semi-structured interviews?) *but have not made any changes to the manuscript. Since you have developed your semi-structured guide through a systematic process, it would greatly enhance your manuscript if you would summarise that process in 4-5 sentences in the methods section. Once you have done that I would be happy to approve your submission for publication.

We look forward to receiving your revised manuscript.

Kind regards,

Abhijit Nadkarni

Academic Editor
---

## [Editor Report · Decision Letter 9]

5 Dec 2024

Coping Mechanisms for Students with Mental Disorders: An Exploratory Qualitative Study at Busitema University’s Mbale and Busia Campuses

PGPH-D-23-01027R9

Dear Ms Kawala,

We are pleased to inform you that your manuscript 'Coping Mechanisms for Students with Mental Disorders: An Exploratory Qualitative Study at Busitema University’s Mbale and Busia Campuses' has been provisionally accepted for publication in PLOS Global Public Health.

Best regards,

Abhijit Nadkarni

Academic Editor